# Profile of a Modern Hunter and the Socio-Economic Significance of Hunting in Poland as Compared to European Data

Krzysztof Kupren *[ID] and Anna Hakuć-Błażowska

Department of Tourism, Recreation and Ecology, Faculty of Geoengineering,
University of Warmia and Mazury in Olsztyn, Oczapowskiego 5, 10-719 Olsztyn, Poland;
hakuc.blazowska@uwm.edu.pl
* Correspondence: krzysztof.kupren@uwm.edu.pl; Tel.: +48-89-524-56-04

**Abstract:** Hunting is a unique form of activity in rural areas with a high proportion of forest areas, which involves nature conservation and meets social needs for recreation and the preservation of traditions while being an important part of economic activity. The presented study results, based on a literature review and questionnaire surveys conducted among hunters associated in hunting clubs in the north-eastern part of Poland, provide the basis for a discussion on the socio-economic significance of hunting, both in the country and throughout the European continent. Based on the results presented in the paper, it can be concluded that the number and density of hunters differ in individual countries. Moreover, hunting is practised in Europe by almost 7 million people, of which 127,000 are in Poland, and is a typical male activity. Most hunters in Poland and other European countries are professionally active inhabitants of rural areas, aged approximately 50 years, with several years of shooting experience and an income exceeding average values. Hunting is an important part of socio-economic activities, particularly in rural areas. It is estimated that in the EU alone, hunting can be worth approximately EUR 16 billion, and creates 100–120 thousand jobs. The most recent results of studies conducted in certain EU countries and the wide range of services provided by the hunting sector indicate that these values may be considerably higher. Regarding Poland, despite the centralised game resource management system, there are no extensive studies of the economic significance of hunting, and the official data are limited to a few basic indices related to hunting statistics. As indicated by the study results presented in this paper, in Poland, hunting-related expenditures are clearly lower than the European average and, thus, the economic significance of hunting is relatively low in this country. Despite this, it is a hunting community that, as a result of the adopted system solutions, is responsible for the functioning of reasonable game management while significantly affecting the management of the vast majority of rural areas.

**Keywords:** rural development; hunting grounds; rural areas

## 1. Introduction

### 1.1. Legal Aspects and Hunting Management in the EU and Poland

Over the centuries, hunting has undergone major shifts. Gradually, it ceased to be the primary source of food supply and became a way of spending free time [1,2]. Currently, in Poland and in the majority of European countries, hunting is a form of nature conservation aimed primarily at adapting the wild animal population to a habitat being constantly changed by humans. Hunting also aims to satisfy social needs regarding the maintenance of traditions and the propagation of hunting ethics and culture [3–5].

Although hunting raises a lot of legal and ethical controversies in many social circles (particularly as regards trophy hunting), and there is no convincing evidence that recreational hunting contributes to sustainable conservation tasks in each case [6–9], it should be noted that in all European countries, irrespective of the motivation and acceptable methods,

it is a legal way to harvest wild natural resources. The legal framework for hunting in the European Union is rather complex. There are a number of legal documents (directives) which, in many cases, are the result of international agreements and the acts derived from them (regulations and decisions) that affect the internal law of each participating EU country. They primarily govern the rules for hunting management and the hunting methods for the sustainable (reasonable) use of natural resources. This primarily applies to the implementation of the so-called Nature Directives (the "Birds" and "Habitat" Directives [10,11]) to manage populations at a level that does not threaten the normal development of game animals, particularly protected species. However, there is no common EU law to govern the common game management of all the EU countries at the lowest level, and there is still the internal issue of the implementation of justified deviations from the regulations issued in each Member State (this applies, e.g., to the list of huntable species or the hunting period) [5]. Nevertheless, modern hunting rules in the European Union are, for the most part, based on the approach set out in the European Charter on Hunting and Biodiversity by the Council of Europe in Strasbourg on 26–29 November 2007 [3].

The above-described situation is currently the case in Poland. Sustainable wild animal population management is achieved in Poland through the application of a centralised hunting model which, compared to the different solutions in individual European countries, enables a high level of hunting activity coordination [5,12,13]. Since game animals belong to the State Treasury, legal regulations concerning game management, defined as an activity in the field of protection, breeding, and harvesting of game (Article 4(1) of the hunting law [4], have been established at the central level (currently, the Ministry of Climate and Environment). Moreover, the vast majority of hunting districts in which game management is pursued, excluding the State Forests' game animal breeding centres, are administered by a single social organisation that joins together hunters, i.e., the Polish Hunting Association, and its constituent hunting clubs, which ensures system coherence throughout the country. This also streamlines the system for compiling hunting statistics and transferring data from hunting districts to higher management levels. Game animals are harvested in accordance with hunting plans (both current, i.e., annual and multiannual) that are developed in detail, reviewed, and approved. Illegal shooting and poaching are prevented by the State Hunting Guard. The Polish Hunting Association and its constituent hunting clubs that associate hunters conduct and fund their activities according to the Association's statutes themselves. In addition to the above-mentioned activities, they also include those aimed at improving the living conditions of animals, i.e., wildlife food plots, buffer plots, meadow reconstruction and mowing, supplementary winter feeding, etc. A major activity is the payment of compensation for damage caused by wild animals. The system has been in place for many years and appears to have been well-organised for most of that period [4,13,14]. However, in recent years, due to the detection of multiple uncertainties (e.g., incorrect estimation of game animals and financial ambiguities), it has been subject to stricter controls by the State institutions [12,15,16].

It should be noted that the most significant difference in game management between most EU countries and Poland is the inseparability of land ownership rights and the right to exercise hunting. If a landowner is entitled to exercise hunting and wants to exercise it on their own land, they can do so, but they also have the option of leasing this right to third parties for remuneration. In both cases, it is the owner that reaps the full benefits of the land they own. The amount of game to be harvested, i.e., the number of animals culled (harvest permits), is usually determined by external bodies that monitor wildlife welfare. As in Poland, lease agreements in most European countries are multiannual; the only difference is that in Poland (but also, for example, in Hungary and Italy), it is not the landowner that benefits from hunting [5,12,13].

*1.2. Social and Economic Significance of Hunting Worldwide*

According to the data provided by the largest hunting organisation in Europe, i.e., the European Federation for Hunting and Conservation (FACE), whose members are national



hunting associations from 37 European countries, including the EU-28, there are currently over seven million hunters in Europe, which makes is the second-largest formally organised hunting population, after the United States of America [17]. The numbers and densities of hunters vary from country to country and even from region to region, which often reflects local hunting traditions, land uses or political circumstances. Consequently, the hunting community represents a diverse group of various social and cultural circles that combines multiple notions and values. In general, a passion for nature and hunting motivates hunters and hunting communities to take a proactive approach to nature conservation. Hunting in the EU alone is estimated to contribute to the management of over 65 per cent of rural areas. It takes place in cooperation with landowners, farmers, foresters and other stakeholders, thus creating an extensive social network involved in nature and landscape management [18].

Hunting is an important socio-economic activity, particularly in rural areas. Recent research reports suggest that in the EU alone, hunting is worth approximately EUR 16 billion [18]. A detailed analysis of hunting expenditures in North America demonstrated an even greater significance and financial contribution of almost 30 billion to the local and national economy. Hunters provide financial support by creating thousands of jobs directly related to the production and sale of goods and services intended to meet their needs. In addition, the expenditures accompanying hunting trips benefit hundreds of thousands of people employed in local shops, restaurants and hotels [19]. Apart from direct expenditures and the creation of jobs, hunting has an additional economic value. The revenue generated from excise taxes imposed on hunting and equipment and from licence fees, support nature conservation and its sustainable management [19].

The above-mentioned data concerning the economic impact of hunting in Europe are estimates. This is due to the fact that individual EU countries are not obliged to draw up and report such lists. Due to having different hunting organisation systems in place, they acquire information in different manners and report differently on game management [5,12,14,18,20].

In view of the above-mentioned differences in the organisation of game management in individual countries, no mechanisms enabling a precise assessment of the value of hunting and its contribution to the EU economy have yet been conducted. The situation is similar as regards the characteristics of hunters. The available socio-economic data concerning hunters in individual European countries are, in most cases, very general or several years old. A good example is the data provided on FACE websites, which, in most cases, date back to the end of the 20th and the beginning of the 21st century, and are most often limited to the presentation of the number of hunters in individual countries as well as their gender [21]. Only a few of the largest hunting associations, e.g., in Spain, Germany, France and the United Kingdom, in recent years have conducted reliable sociological studies on hunters, also determining the scale of their expenditures, and published them on their websites [22–25].

In Poland, thanks to the centralised hunting model, there are accurate data available on hunting statistics, including the value of harvested animals and the compensation for hunting damage. On the other hand, there are no extensive, reliable data on the economic value of hunting. The situation is similar for the characteristics of hunters.

The determination of the economic consequences of hunting, and the identification of groups of users involved in the management of wildlife resources is critically important in the context of management improvements and political decisions related to hunting and biodiversity conservation [18,26,27]. The socio-economic benefits generated by hunting are becoming even more important in the context of rural development. In many under-industrialised regions of the world, tourism, including hunting tourism, is an important form of activity that contributes to an improvement in the living conditions of the local population [28,29].

The aim of the study is to attempt to show the socio-economic profile of the hunter as well as the social and economic dimensions of hunting in Poland against the background of this social phenomenon in other parts of Europe.

## 2. Materials and Methods

Given the selective and, in most cases, outdated statistical and literature data concerning the characteristics of hunters in Poland, the acquisition of data describing this social group was based on the diagnostic survey method. The study involved hunters hunting in different parts of north-eastern Poland. The area selected for the study, comprising several dozen hunting districts of Warmińsko-Mazurskie Voivodeship, is characterised by very favourable conditions for hunting (the percentage of agricultural land in a particular area is about 40%, and that of forests is about 50%), and the percentage ratio of hunters to the general population is one of the highest in Poland [30] (Figure 1).

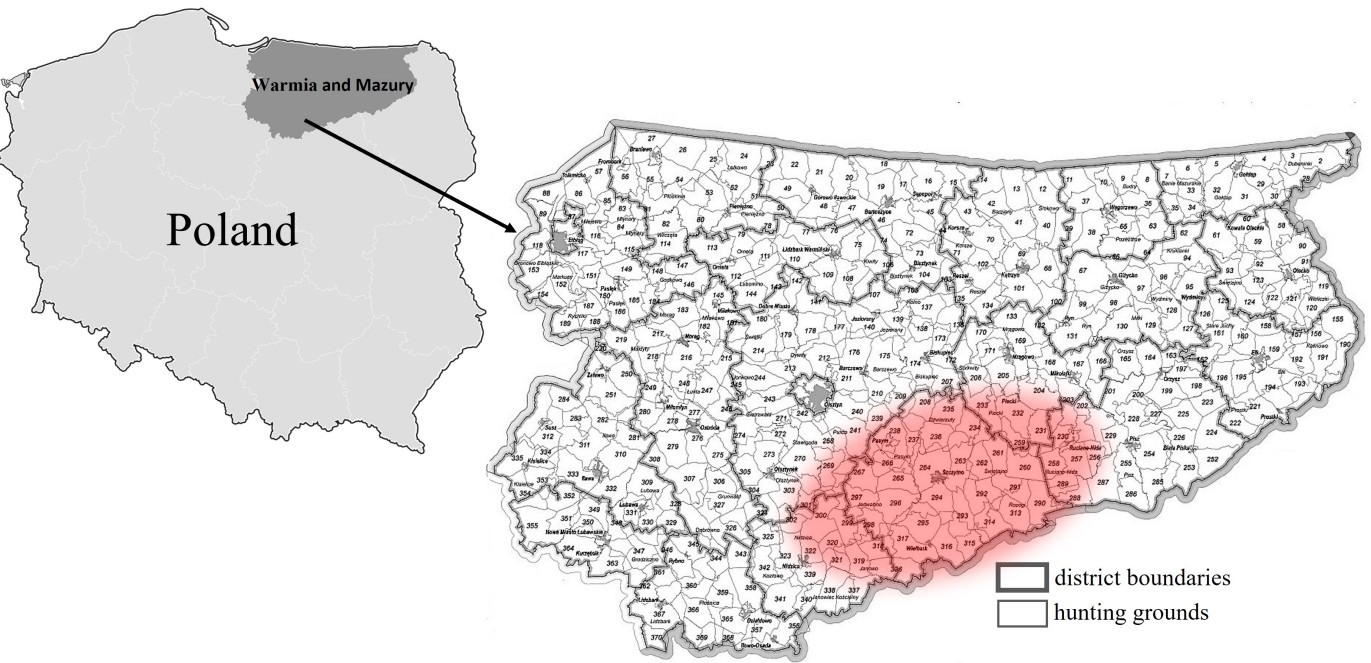

**Figure 1.** The area covered by the diagnostic survey (red area).

Structured interviews with hunters were conducted from January to March 2019. The study was conducted alongside hunting meetings and events held in the field. Participation in the study was voluntary and anonymous, and the only prerequisite was to conduct hunting activity. The interviews began with a short description explaining the goals and background of the study. The questions asked were intended to determine the demographic and sociological profile of the group under study (i.e., gender, age, place of residence, educational background, professional status and hunting experience), the level of the income earned and the annual expenditures on hunting-related purposes. The thematic scope of the questions and the study performance method were drawn from studies on angling after adjusting them for hunting [31,32].

An important supplement of the study was an analysis of statistical and literature data dedicated to various social aspects (including changes in the number of hunters) and economic aspects (hunting expenditures, the share in the GDP and employment) of hunting in Europe. For this reason, the most recent publicly available reports, studies and articles on hunting were used. What was also helpful was an analysis of websites of hunting associations in individual European countries, primarily the information gathered and made public by the FACE. The presented review and synthesis of literature data concerning the social and economic aspects of the hunting effects were not aimed at their detailed

analysis in individual countries, but instead at demonstrating the most recent empirical evidence on the scale of economic effects and the significance of hunting for the economy of Poland and other EU countries.

## 3. Results and Discussion

### 3.1. The Number of Hunters: Current Status and Trends

The hunting population size and its changes over time are important for the management of wild animal populations. According to official statistics, there are 127,426 hunters in Poland (i.e., 1.9% of all the hunters registered in the EU), associated in 2705 hunting clubs that lease 4622 field and forest hunting districts covering an area of 25,216.5 thousand [30]. Despite the vast areas in which hunting is permitted (approximately 82% of the area of Poland), hunters account only for 0.32% of the population, i.e., there is one hunter for every 313 citizens living in Poland. In the European Union, in which approximately 5.9 million hunters are registered, these values are clearly higher and amount to 1.14% and 87, respectively [30,33].

Recent statistical data and the data published by the FACE indicate that the popularity of hunting, measured by the ratio of hunters to the general population, is the greatest in Scandinavian countries and in southern countries with Romance culture [33–35]. In Scandinavia, where the hunters-to-inhabitants ratio is below 1:36, hunting has an eminently recreational and somewhat trapper-like character. In the latter group of countries (including France, Spain, Portugal and Greece), this ratio is slightly higher yet still below the average for the EU (1:87). These are countries with a low intensity of game management, an organisational system that is often licence-based and district-free and places high hunting pressure on the fauna. Poland is among other countries such as Switzerland, Germany, Romania, Belgium and the Netherlands with a very high inhabitants-to-hunters ratio (above 200). These are countries with a long tradition of hunting based on the feudal model [33,34].

In recent decades, a downward trend in the hunting population has been noticeable in Europe. The most pronounced drop (several percent) was observed in the 1990s and particularly concerned certain countries in southern Europe, where hunting was very popular (e.g., Spain, France and Italy). This trend clearly slowed down after 2006. Since then, in most cases, the stabilisation of the trend or an increase in the number of hunters could be observed [36–38]. A good example of a slow, yet systematic increase in the number of hunters in the last decade are the Central European countries, particularly Germany, Austria and Poland [24,39,40]. The hunting populations in individual countries and their changes over time are a result of many factors. These are primarily determined by the legal and administrative rules, the natural conditions for practising hunting, economic factors (e.g., hunting-related fees) and sociological determinants [18,34,36]. It should be stressed that the public acceptance of any form of recreational hunting, even involving invasive alien species, is currently lower than at any time in the past, with concerns about animal welfare and animal rights being predominant in discussions and ethical considerations about the moral implications of hunting for pleasure. The controversial nature of modern hunting lies in the questioning of the advisability of killing animals and methods that fail to harmonise with current ethical and legal standards regarding slaughter and killing [8,9]. The relatively low popularity of hunting in Poland is largely due to legal, administrative and economic factors. The complex, time-consuming and costly procedures for obtaining hunting permits (long-term and expensive training, and restrictive requirements to obtain a permit for hunting weapons) [13,41].

### 3.2. Sociological Characteristics of Hunters

The available European statistics and literature mainly present the basic demographic and sociological data that characterise hunters. For the most part, they only concern the hunters' gender and age. The study conducted for the purposes of this paper involved 100 hunters and aimed at determining the socio-economic profile of hunters. All study participants were members of local hunting clubs. The most important socio-demographic

data describing the hunters in this study are provided in Table 1. The data reveal that the vast majority of hunters, i.e., more than 95%, are men (Table 1). According to the data presented by the Polish Hunting Association, 97% of hunters in Poland are men [40].

**Table 1.** Socio-demographic profile of hunters from the studied Region of Warmia and Mazury.

| Item | Options | Percentage |
|---|---|---|
| Gender | Female | 5 |
| | Male | 95 |
| Age (years) | <18 | 0 |
| | 18–25 | 15 |
| | 26–40 | 36 |
| | 41–60 | 49 |
| | >60 | 18 |
| Education background | Basic | 4 |
| | Vocational | 18 |
| | Secondary | 48 |
| | Higher | 30 |
| Professional status | Unemployed | 0 |
| | Student | 10 |
| | Active | 68 |
| | Retired | 22 |
| Place of residence | City | 39 |
| | Village | 61 |
| Hunting experience (years) | <5 | 17 |
| | 5–10 | 14 |
| | 11–20 | 24 |
| | 21–30 | 20 |
| | >30 | 25 |
| Total | | 100 |

Studies conducted in other parts of the world also indicate that hunting is a typical male activity, while the percentage of actively hunting women is negligible. A small percentage of female hunters, similar to that provided in Polish data, is found in most other EU countries. Examples are found in Finland and Austria, where approximately 10% of hunters are women [27,39]. Despite the clear male predominance, a few countries have seen a noticeable increase in interest in hunting on the part of women in recent years. This is particularly observed in the countries with the general increase in the number of hunters (e.g., in Germany, Austria and Poland). Every indication is that this phenomenon is mostly contributed to by the promotional activities of hunting associations in which the crucial argument appears to be the environmental benefits provided by hunters [24,39].

For legal reasons, one must be an adult to hunt. The age of almost half of the hunters under study ranged from 40 to 60 years (with an average age of 47 years). According to the Polish Hunting Association data, the average age of hunters in Poland is 52 years [40]. The situation in this respect is similar in other European countries where fifty-year-olds have long been the largest group of hunters [23,24,27].

Similarly to the situation, e.g., in France, Germany or Spain, the largest portion of hunters in the region of Poland under study comprises persons with secondary or higher education and professionally active, while the percentage of students is low. In most cases, hunters are also inhabitants of villages and rural areas (Table 1) [23–25,27]. The results obtained in this study are consistent with the European data, also in terms of the hunters' experience. The available data indicate that the vast majority of hunters have been interested in this activity for several years, and the data from Spain, France and Germany also show that the main reason for their interest in hunting has been family traditions [23,24,27,42].

*3.3. Economic Aspects of Hunting*

Hunters make a contribution to all major sectors of the economy, both directly and indirectly. For example, they compensate farmers for crop damage in the primary sector, purchase equipment from the secondary sector and pay for tourism services in the tertiary sector. As a result of generating these values, and in order to sustain hunting, a certain amount of money and other resources is also reinvested in the conservation or restoration of habitats and wild animal populations [18].

All the available data about the contribution of hunting to the economy refer mostly on hunters' hunting expenditures, acquired using questionnaire surveys. In the 1990s, the data obtained from a few Western European countries indicated that, on average, a single hunter spends approximately an average of EUR 1500 per annum on their hobby [34]. The research conducted at the end of the subsequent decade, this time involving hunters from all EU Member States at that time, showed an amount of EUR 2500 [43]. All expenses related to hunting, e.g., licences, leases, weapons and ammunition, equipment and trips, were considered. However, no social aspects or those related to nature conservation were taken into account. The average value of a hunter's expenditures, extrapolated to the entire hunter population (approximately 6.6 million people), amounts to EUR 16 billion and is the amount most frequently quoted in FACE reports as the economic value of hunting in Europe.

It is worth mentioning that, in addition to the level of an average hunter's expenditures, the above-cited paper [43] also reports on a varied level of expenditure declared by the respondents. They ranged from EUR 700–4300, but without indicating the expenditures in individual countries [43]. The current level can be estimated in individual countries from websites of individual national hunting associations. In-depth studies dedicated to these issues have been conducted in the last decade, inter alia in the United Kingdom, France, Spain and Germany, i.e., in countries where hunting is very popular, or its popularity is on the increase. The average hunter's expenditures in the indicated countries exceed the above-cited European average, yet they do not differ significantly from the upper range limit indicated. For example, the amount is EUR 2800 in France, GBP 2000 in the United Kingdom, and EUR 4340 in Germany [22–25]. The situation is different in Spain, where the expenditures declared by hunters are almost four times greater than the average value (EUR 9649). The data from Spain present the component structure of the expenditures with exceptional accuracy. An interesting fact is that for an average of 27 hunting trips in a year, almost half of the amount of EUR 9649 declared by hunters covers expenditures on transport (including car maintenance), accommodation (including second house maintenance) and food [25].

As for Poland, the official economic hunting-related data list the quantity and values of the procurement of game animals and the compensation paid from the sources of managers or leaseholders of hunting districts for losses in agricultural crops and the damage caused by hunting. In 2020, these values amounted to PLN 108,432,400 (approximately EUR 25 million) and PLN 92,603,200 (approximately EUR 21.5 million), respectively [30]. In contrast, there is no information on the expenditures of hunters alone or on the contribution of hunting to the national economy. Research conducted for the purpose of this study shows that the average annual expenditures of hunters in Poland, including all expenditures related to hunting, amounts to an average of PLN 2702 (approximately EUR 640) (Table 2).

**Table 2.** Average annual expenditures directly related to hunting and monthly gross income of hunters from the studied Region of Warmia and Mazury. Values expressed in PLN (Polish New Złoty).

| Item | Options | Percentage |
|------|---------|------------|
| Expeditures | <1000 | 11 |
| | 1001–2500 | 51 |
| | 2501–5000 | 28 |
| | 5001–7500 | 8 |
| | >7500 | 2 |
| | *Average: 2702* | |
| Monthly gross income | <2500 | 25 |
| | 2501–3500 | 31 |
| | 3501–4500 | 24 |
| | >4500 | 20 |
| Total | | 100 |

When comparing this data with the European data presented above, it can be noted that these values are four times lower than the average and fall just below the lower limit of the expenditure range indicated in 2008 [43]. The observed values and, possibly, the differences in individual countries result from the slightly different (not always recommended) research methodology and are determined by the hunting method and the economic factors, such as the amount of hunting fee or the income earned [18,34]. Regarding the group of hunters from Poland under study, even though the average gross monthly earnings were close to the average level of income in the region under study, they were still a third of the value noted in the EU Table 2 [43,44]. It is not without significance that the declared expenditure level was undoubtedly affected by the fact that the vast majority of the respondents hunted in close proximity to their place of residence. Hunters in this area usually hunt within a radius of up to 15 km from their place of residence, in hunting districts belonging to the hunting club of which they are members. [45]. It is also worth noting that the reported amount of hunters' expenditures in Poland was greater by more than half of that noted for another group harvesting wild animal resources, i.e., anglers [32,46]. In the case of this group, the vast majority of people in all regions of Poland practise their hunting activity on a very local basis, using mainly the local environmental resources. If the above-mentioned hunting expenditures were taken as those reflecting the situation throughout the country, and if they were extrapolated to the entire population of 127,000 hunters, the direct expenditures by hunters in Poland would amount to approximately PLN 343 million (approximately EUR 80 million), i.e., less than 0.5% of the expenditures in Europe, estimated at EUR 16 billion [18].

Moreover, in order to get a realistic picture of the economic significance of hunting as a whole, it would be necessary to consider a much broader set of impacts embedded in hunting activities, in addition to the direct hunting expenditures (i.e., for hunting equipment, trips, game animal maintenance, licences, taxes, trophies, etc.). These include the economic, environmental and cultural effects related to species conservation and management, restoration of habitats and land management provided by hunters. Many of these costs would have to be borne by taxpayers to fund the restoration and management of habitats/species, or compensate landowners for damage caused by game animals in the absence of hunting. Other manifestations of the positive impact of hunters and hunter community (including hunters' families, friends, etc.) include the promotion of culture, heritage, tourism, local economy, welfare and voluntary work in activities related to both habitat and wildlife management. Some of these activities are difficult to measure, and most of them, due to methodological difficulties, have not yet been valued [18,27,34].

In recent years, a few European countries have made an attempt to take different activities performed voluntarily by hunters into account in the most characteristic economic indices, e.g., the share in the GDP or the creation of jobs, both those directly and indirectly dependent on hunting. Since some data (based primarily on the direct benefits

generated by hunting) suggest that in Europe, one job is generated by 65 hunters, it can be approximated that European hunters support between 100,000 and 120,000 jobs [34]. The most recent research conducted in countries with the greatest numbers of hunters indicates a considerably greater impact of hunting on the economy than that mentioned above. It is estimated that in 2014, in the United Kingdom alone, 600,000 hunters and target shooters spent an estimated GBP 2.5 billion on goods and services, and the total gross value added, related to sport shooting, is estimated at GBP 2 billion (approximately EUR 2.6 billion). It was calculated that this community also creates 74,000 jobs, of which half (35,000) are directly dependent on hunting. Accommodation and catering are the sectors with the largest percentages of these jobs. Nature conservation-related works alone, involving approximately 3.9 million days of conservation work, correspond to 16,000 jobs [20]. In Italy, the annual total costs incurred by 850,000 official hunters is estimated at €3.26 billion, and hunting and shooting further create a little less than 43,000 jobs in total [47]. Similar figures are generated by the French hunting sector. In addition to a turnover of EUR 3.9 billion per annum and EUR 2.3 billion in value added to the national economy (GDP), the activities of 1.1 million hunters guarantee 28,000 permanent jobs, and volunteer works (including activities related to the management of natural habitats and wildlife) create a further 57,000 full-time jobs [23]. Even higher figures are noted for Spain, which is similar to Poland in terms of the area and population. These figures indicate that just over 713,000 people with hunting licences generate almost 1% (187,000) of all jobs [25].

## 4. Conclusions

According to the data presented in this study, despite the increased interest in hunting in recent years, Poland is among the countries with the relatively smallest number of hunters in the EU. Despite the various levels of hunting popularity throughout Europe, the socio-demographic profile of hunters appears to be very similar. The only factor that appears to clearly differentiate hunters in Poland from hunters from other Western European countries is the relatively low declared expenditures for hunting. This is most probably due to economic factors, and may be linked to the hunting method. Despite the centralised and unified system of hunting management throughout Poland, no attempts have yet been made to determine the economic significance of hunting and its impact on the economy. The results of these studies, even though methodologically different, are available in other European countries. Apart from the indisputable social and economic benefits, it is also important to remember the extremely important ecosystem services provided by hunters, which very often are not known to society, and their final valuation is very difficult.

Nevertheless, based on the data gathered, it can be concluded that without the financial and social support provided by hunters, modern wildlife management in European countries would undoubtedly be very difficult. This social dimension is particularly important in Poland. This is connected with the way of pursuing game management, in which hunters who are members of hunting clubs belonging to the Polish Hunting Association administer the hunting grounds themselves, thus having a great influence on nature management and, in a broader dimension, on the functioning of rural areas as well.

**Author Contributions:** Conceptualization: K.K.; methodology: K.K.; formal analysis: K.K. and A.H.-B.; investigation: K.K.; resources:, K.K.; data curation: K.K.; writing—original draft preparation, K.K. and A.H.-B.; writing—review and editing: K.K.; A.H.-B.; supervision: K.K. and A.H.-B. All authors have read and agreed to the published version of the manuscript.

**Funding:** This research received no external funding.

**Institutional Review Board Statement:** The study was approved by the Scientific Research Ethics Committee of the University of Warmia and Mazury in Olsztyn (DECISION No 6/2018 of the SCIENTIFIC RESEARCH ETHICS COMMITTEE).

**Informed Consent Statement:** Not applicable.

**Data Availability Statement:** Not applicable.

**Acknowledgments:** The authors would like to thank Dominika Kordek and Aneta Omelan for their technical support in the field research and the three anonymous reviewers for their insightful suggestions and careful reading of the manuscript.

**Conflicts of Interest:** The authors declare no conflict of interest.

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
