# Peer review of "Profile of a Modern Hunter and the Socio-Economic Significance of Hunting in Poland as Compared to European Data"

_land, doi:10.3390/land10111178_

Round 1
Reviewer 1 Report
The paper is interesting from a socio-economic point of view
there is no deepening in Italy, where hunting is an important social phenomenon; it should then be considered the great opposition that exists on the part of environmentalists in many European countries
Author Response
The paper is interesting from a socio-economic point of view
there is no deepening in Italy, where hunting is an important social phenomenon; it should then be considered the great opposition that exists on the part of environmentalists in many European countries
Firstly, we would like to thank you very much for your valuable comments and constructive reviews for improving our manuscript. Please find herewith the response to your comments “point-by-point” along with a marked‐up-revised version of our manuscript showing the changes required according to your comments, newly added paragraphs, amended text, and updated references in “red-colored text”. We believe that we have addressed your comments to an appropriate degree.
Response:
We agree that Italy is a very important country from the point of view of European hunting. There are references to Italy in several places in the article. For example, it concerns the number of hunters. In the part on the economic importance of hunting, data from Italy have also been added.
(please see lines 216-219; 396-398).
At the beginning of the introduction and in the discussion, the ethical and environmental aspects of hunting have also been extended or added (please see lines 40-43; 227-233). Related citations have also been added:
- Varner, G. E. In Nature's Interests?: Interests, Animal Rights, and Environmental Ethics, Oxford University Press, New York, USA, 2002;. 164 pp
- Di Minin, E.; Clements, H.S.; Correia, R.A.; Cortes Capano, G.; Fink, C.; Haukka, A.; Hausmann, A.; Kulkarni, R.; Bradshaw, C. Consequences of recreational hunting for biodiversity conservation and livelihoods. One Earth, 4(2), 2021, 238-253. https://doi.org/10.1016/j.oneear.2021.01.014
Additional comments and authors' changes:
Changes in the figures introduced in the chapter 3.1 are due to the fact that the previous ones applied to the whole of Europe and the current ones refer to the EU countries (as mentioned in the text)
(please see lines 202-210)
Reviewer 2 Report
Review of land-1418040
Profile of a Modern Hunter and the Socio-Economic Significance of Hunting in Poland as Compared to European Data
The introduction, situating hunting in Poland into the European context and the discussion of the way it is practiced and managed in Poland are well developed
The same applies to section 1.2
What is not well developed if not missing altogether, though, is less a narrative of the situation, but a contextualising and framing in term so of the existing literature on the topic in other areas and other countries.
Table 1: if the total population is 100 hunters, then there is no need to have n and % columns One will suffice.
The methodology section does not explain the process how these 100 hunters were selected / chosen /identified. This needs to be addressed.
The discussion and conclusions are adequate
Of major concern is the following:
It is unclear whether the authors obtained permission from the ethics review board of their institution for their structured interviews. In this day and age such should be stated. Not doing so conjures up the spectre of rogue science. If no ethics approval was obtained, the paper must be rejected because any publication would undermine the standing of LAND
.
MINOR ISSUES
The paper needs athrough edit by a native English speaker for spelling and grammar
See Table 1: 'Proffesional status' '‘Hunting expirence ' 'Willage'
Line 294 closimg bracket is missing
Author Response
Profile of a Modern Hunter and the Socio-Economic Significance of Hunting in Poland as Compared to European Data
The introduction, situating hunting in Poland into the European context and the discussion of the way it is practiced and managed in Poland are well developed
The same applies to section 1.2
What is not well developed if not missing altogether, though, is less a narrative of the situation, but a contextualising and framing in term so of the existing literature on the topic in other areas and other countries.
Firstly, we would like to thank you very much for your valuable comments and constructive reviews for improving our manuscript. Please find herewith the response to your comments “point-by-point” along with a marked‐up-revised version of our manuscript showing the changes required according to your comments, newly added paragraphs, amended text, and updated references in “red-colored text”. We believe that we have addressed your comments to an appropriate degree.
Response:
We agree that the information on hunting on hunting management in other countries has been treated very vaguely. This is due to the very different approach in different countries, their description would take a large part of the manuscript. In the new version, in addition to the extended citation, there is information about the most important differences between Poland and other countries. In our opinion, this form is sufficient and will not distract too much from the essence of the considerations contained in the manuscript. Please see lines 87-100.
At the beginning of the introduction and in the discussion, the ethical and environmental aspects of hunting have also been extended or added (please see lines 40-43; 227-233). Related citations have also been added:
- Varner, G. E. In Nature's Interests?: Interests, Animal Rights, and Environmental Ethics, Oxford University Press, New York, USA, 2002;. 164 pp
- Di Minin, E.; Clements, H.S.; Correia, R.A.; Cortes Capano, G.; Fink, C.; Haukka, A.; Hausmann, A.; Kulkarni, R.; Bradshaw, C. Consequences of recreational hunting for biodiversity conservation and livelihoods. One Earth, 4(2), 2021, 238-253. https://doi.org/10.1016/j.oneear.2021.01.014
Table 1: if the total population is 100 hunters, then there is no need to have n and % columns One will suffice.
Response:
The tables have been changed according to reviewer suggestion.
The methodology section does not explain the process how these 100 hunters were selected / chosen /identified. This needs to be addressed.
Response:
The methodological section has been expanded the most important information regarding the identification of the hunters and how to conduct the interviews has been included in the new version of the manuscript (please see lines 172-176).
The thematic scope of the questions and the study performance method were drawn from cited studies [31,32].
The discussion and conclusions are adequate
Of major concern is the following:
It is unclear whether the authors obtained permission from the ethics review board of their institution for their structured interviews. In this day and age such should be stated. Not doing so conjures up the spectre of rogue science. If no ethics approval was obtained, the paper must be rejected because any publication would undermine the standing of LAND
Response:
We forgot to include the ethics committee approval. Of course The study (structured interviews) was conducted following the ethical recommendations of the Scientific Research Ethics Committee of the University of Warmia and Mazury in Olsztyn (DECISION No 6/2018 of the SCIENTIFIC RESEARCH ETHICS COMMITTEE) The relevant authors' statement will be included in the new version of the manuscript:
The study was approved by the Scientific Research Ethics Committee of the University of Warmia and Mazury in Olsztyn (DECISION No 6/2018 of the SCIENTIFIC RESEARCH ETHICS COMMITTEE)
MINOR ISSUES
The paper needs athrough edit by a native English speaker for spelling and grammar
See Table 1: 'Proffesional status' '‘Hunting expirence ' 'Willage'
Line 294 closimg bracket is missing
Response:
The corrected version of the manuscript was sent to a Native English speaker for English editing and proof.
All remarks and suggestions were provided in the text
Reviewer 3 Report
Although the manuscript’s ambitions are modest, the author/s take a clear narrow question on the importance of hunting in the Polish versus European contexts, using both survey and secondary data. The comparisons are well explicated and applied with the findings intuitively argued towards reasonable implications.
Author Response
Although the manuscript’s ambitions are modest, the author/s take a clear narrow question on the importance of hunting in the Polish versus European contexts, using both survey and secondary data. The comparisons are well explicated and applied with the findings intuitively argued towards reasonable implications.
Firstly, we would like to thank you very much for your valuable comments and constructive reviews for improving our manuscript. Please find herewith the response to your comments “point-by-point” along with a marked‐up-revised version of our manuscript showing the changes required according to your comments, newly added paragraphs, amended text, and updated references in “red-colored text”. We believe that we have addressed your comments to an appropriate degree.
Additional comments and authors' changes:
Changes in the figures introduced in the chapter 3.1 are due to the fact that the previous ones applied to the whole of Europe and the current ones refer to the EU countries (as mentioned in the text)
(please see lines 202-210)
The corrected version of the manuscript was sent to a Native English speaker for English editing and proof.
Round 2
Reviewer 2 Report
The authors have adequately addressed the issues raised in my review. Thank you
Author Response
Thank you very much for your valuable comments and constructive reviews for improving our manuscript.